# Root Colonization by Fungal Entomopathogen Systemically Primes Belowground Plant Defense against Cabbage Root Fly

**DOI:** 10.3390/jof8090969

**Published:** 2022-09-16

**Authors:** Catalina Posada-Vergara, Katharina Lohaus, Mohammad Alhussein, Stefan Vidal, Michael Rostás

**Affiliations:** 1Agricultural Entomology, Department of Crop Sciences, University of Göttingen, Grisebachstr 6, 37077 Göttingen, Germany; 2Molecular Phytopathology and Mycotoxin Research, Department of Crop Sciences, University of Göttingen, Grisebachstr 6, 37077 Göttingen, Germany

**Keywords:** endophytic entomopathogenic fungi, *Delia radicum*, *Metarhizium brunneum*, *Brassica napus*, plant-fungal-insect interactions, split-root, glucosinolates, jasmonic acid

## Abstract

Entomopathogenic fungi infect insects via spores but also live inside plant tissues as endophytes. Frequently, colonization by entomopathogens provides plants with increased resistance against insects, but the mechanisms are little understood. This study investigated direct, local, and systemic root-mediated interactions between isolates of the fungus *Metarhizium brunneum* and larvae of the cabbage root fly (CRF) *Delia radicum* attacking *Brassica napus* plants. All fungal isolates infected CRF when conidia were present in the soil, leading to 43–93% mortality. Locally, root-associated *M. brunneum* isolates reduced herbivore damage by 10–20% and in three out of five isolates caused significant insect mortality due to plant-mediated and/or direct effects. A split-root experiment with isolate Gd12 also demonstrated systemic plant resistance with significantly reduced root collar damage by CRF. LC-MS analyses showed that fungal root colonization did not induce changes in phytohormones, while herbivory increased jasmonic acid (JA) and glucosinolate concentrations. Proteinase inhibitor gene expression was also increased. Fungal colonization, however, primed herbivore-induced JA and the expression of the JA-responsive plant defensin 1.2 (PDF1.2) gene. We conclude that root-associated *M. brunneum* benefits plant health through multiple mechanisms, such as the direct infection of insects, as well as the local and systemic priming of the JA pathway.

## 1. Introduction

Species in the genus *Metarhizium* (Ascomycota, Clavicipitaceae) and *Beauveria* (Ascomycota, Cordycipitaceae) are widespread fungal pathogens of insects. These entomopathogens proliferate in the rhizosphere but can also live as endophytes inside plants, where they mainly colonize the root tissue [1,2,3,4]. Due to these characteristics, entomopathogens are potential candidates for the control of soil-borne insect pests. Not only can the fungi infect insects that approach the root system in search for food, but as endophytes they may also induce changes in the plant that negatively affect insect performance [5,6]. The mechanisms leading to reduced feeding or enhanced mortality in insects are not well understood, but recent studies have suggested that endophytic entomopathogenic fungi (EEF) may induce systemic resistance (ISR) in plants [7,8]. ISR is regulated by jasmonic acid (JA) and ethylene signaling and can be elicited by a variety of beneficial microbes, including plant growth promoting rhizobacteria and non-pathogenic fungi [9,10,11,12]. However, apart from JA and ethylene, several beneficial microorganisms also induce salicylic acid (SA)-dependent resistance [12].

Plant responses to EEF vary considerably and happen to be isolate and species specific, ranging from suppression to the activation of hormonal pathways and defense-related metabolites. For instance, in *Arabidopsis thaliana* plants two endophytic strains of *Beauveria bassiana* showed significant differences in the upregulation of genes involved in JA and SA signaling, phytoalexin synthesis, and other plant defense pathways [13]. Likewise, a study with six different *B. bassiana* isolates demonstrated the differential activation of *Nicotiana benthamiana* genes involved in JA or SA signaling [14]. Comparing two different EEF species, Rasool et al. [15] reported that *B. bassiana* and *M. robertsii* increased the production of defense-related benzoxazinoids in wheat and flavonoids in bean plants, while *M. brunneum* reduced the concentration of the same metabolites.

Apart from inducing plant responses, beneficial microorganisms can also mediate systemic resistance through priming [11]. Resistance conferred through priming is characterized by increased sensitivity to JA and ethylene due to microbe colonization, and in some cases involves the SA pathway [16,17]. Primed plants show faster and/or stronger activation of cellular defenses when a biotic or abiotic stressor is perceived as a second trigger [11,18,19,20]. Priming has also been demonstrated in EEF recently. For example, *M. brunneum* primed cauliflower plants for increased myrosinase activity in response to herbivory by *Plutella xylostella* [8]. Myrosinase is a crucial enzyme in the glucosinolate defense system of Brassicaceae.

So far, root-colonizing EEF have been shown to affect herbivorous insects systemically when feeding on aboveground plant parts [8,15]. Whether similar indirect interactions also exist belowground, where EEF induce or prime defenses against herbivores feeding on uncolonized root tissues, has not been evaluated so far. However, split-root systems have been used to demonstrate the systemic effects of other root-associated fungi such as *Fusarium oxysporum*, *Pochonia chlamydosporia*, or *Trichoderma* spp. on phytopathogenic nematodes [16,21,22,23,24].

The cabbage root fly (CRF) *Delia radicum* (Diptera, Anthomyiidae) is an economically important pest of several Brassicaceae crops in the temperate zone [25]. The female fly lays its eggs into the soil, close to the base of the plant. After hatching, the first instar larvae feed on root hairs and then migrate to the tap root where they feed on periderm, phloem, and parenchyma tissue [26]. Wounded roots are entry points for soil-borne diseases [27] and when infestation rates are high, inflicted root damage can lead to decreased yields [28]. 

Previous studies have shown that CRF larvae are susceptible to *Metarhizium* spp. [3,29,30,31] and that plants inoculated with *M. anisopliae* were less damaged by the insect [32]. As with plant responses, isolate-specific differences can be found in pathogenicity towards insects [31,33,34] and in the ability of *Metarhizium* spp. to form associations with plants [35,36]. Hence, the first objective of this study was to compare the pathogenicity of different isolates of *M. brunneum* in the presence and absence of *Brassica napus* plants and, furthermore, to establish their rhizosphere competence and endophytic capabilities. The second aim was to elucidate whether any reduction in root damage must be attributed entirely to *M. brunneum* infecting CRF larvae in the rhizosphere (direct effects) or, additionally, to fungus-mediated plant responses (indirect effects). To this end, a split-root setup was used that enabled differentiation between local and systemic responses [16,21,37,38,39]. The results suggest that endophytic *M. brunneum* can protect *B. napus* plants in both ways: by infecting CRF larvae in the rhizosphere and by priming for jasmonic acid-dependent plant responses locally and systemically. 

## 2. Materials and Methods

### 2.1. Study System

Oilseed rape plants (*B. napus* var. Penn) were grown from seeds provided by Norddeutsche Pflanzenzucht (Hans-Georg Lembke KG, Holtsee, Germany) in a non-sterile soil mix, consisting of field loam, sand, and vermiculite in a ratio of 2:1:0.25 (by volume). Plants were grown in a rearing room at 19 °C, 60–70% RH and a 16:8 h (L:D) photoperiod with high pressure sodium lamps. All in vitro experiments were incubated in a growth cabinet (Mytron, Bio- und Solartechnik GmbH, Heilbad Heiligenstadt, Germany) at 19 °C and 65% RH in darkness if not specified otherwise.

Isolates of *M. brunneum* were obtained from the in-house collection of the Division of Agricultural Entomology (see Appendix A). To obtain spore suspension for all experiments, isolates were grown on potato dextrose agar (PDA) (Carl Roth GmbH, Karlsruhe, Germany) at 23 °C for 14 days. The conidia were removed from the hyphae by gently scraping the sporulating colony with a sterile glass slide and the conidia were suspended in 20 mL of 0.1% Tween 80 (Carl Roth GmbH, Karlsruhe, Germany). The suspension was filtered through a plastic gauze and adjusted to a final concentration of 1 × 10^7^ mL^−1^. Spore viability was assessed before each experiment. 

CRFs were reared under controlled conditions at 20 °C, 60–80% RH and a 16:8 h (L:D) photoperiod. Adults were kept in rearing cages (30 × 30 × 30 cm, BugDorm, Megaview Science Co., Ltd., Taichung, Taiwan) and fed with a diet of dry food, consisting of dextrose, skim milk powder, soy flour, and brewer’s yeast in a ratio of 10:10:1:1 (by weight) and wet food consisting of honey, soy flour, and brewer’s yeast in a ratio of 5:1:1 (by volume). To stimulate oviposition, a glass Petri plate filled with washed coarse sand and a piece of rutabaga (*B. napus* L. var. *napobrassica*) was placed inside the cage for 24 h. The eggs were extracted from the substrate by flotation. To obtain larvae, 100 eggs were placed on rutabaga slices (300 g) that were previously scratched to facilitate penetration by the neonate larvae. Eggs were then incubated in plastic boxes (1 L) filled with washed sand (2–5 mm). After 21 days, third instar larvae (L_3_) were carefully extracted from the rutabaga. All eggs and larvae were used in the following 2 h after flotation and extraction, respectively. 

### 2.2. Susceptibility of CRF to Different M. brunneum Isolates in Substrate without the Plant

The pathogenicity of *M. brunneum* was evaluated on L_3_ larvae since this is the stage that leaves the root tissue to pupate in the soil. The experimental unit consisted of 35 mm black film canisters filled with either 20 cm^3^ of sterile silica sand or with the non-sterile soil mix. The substrate was inoculated by adding 1 mL of spore suspension on top of the substrate. Controls received 1 mL of 0.1% Tween 80. Three L_3_ larvae were then released into each container, making sure the larvae buried into the substrate. The film canisters were covered with Parafilm M and placed in the growth chamber. Daily emergence of adult flies was evaluated from day 16 to 21 until no more adults emerged. The remaining pupae were recovered by flotation, surface-sterilized with 70% ethanol for 30 s, and rinsed three times in sterile water. The mycosis of the adults and remaining pupae was further evaluated by placing them in Petri dishes lined with wet filter paper and incubated at 23 °C and 65% RH in darkness. Plates were observed daily for 10 days. The experimental setup consisted of six treatments (five fungal isolates, one control), two substrate types with 12 replicate plants per treatment. All treatments were kept completely randomized in the climatic chamber. Variables measured were the number of adult flies that emerged and the total mortality of CRF (%) defined as
$$Motality\;\%\;=\frac{\;\left(mycosed\;larvae\right)+\left(mycosed\;and\;dead\;pupae\right)+\left(mycosed\;flies\right)}{introduced\;larvae}\times 100$$

Total mortality was also used to calculate Abbott’s corrected mortality [40].

### 2.3. Rhizosphere Competence, Endophytism, and Plant Protection of M. brunneum Isolates

Rhizosphere competence and the endophytic ability of five *M. brunneum* isolates in oilseed rape plants were evaluated, and their pathogenicity towards CRF was assessed (for isolates, see Appendix A). Seeds were surface sterilized in 70% ethanol for 1 min and 2% sodium hypochlorite (Carl Roth GmbH, Karlsruhe, Germany) for 5 min. Seeds were then rinsed three times and sown in sterile silica sand. One week after sowing, seedlings were inoculated by root dipping in spore suspension for 30 min and then transplanted to square pots (13 × 13 cm, 2 L) filled with non-sterile soil mix. Control seedlings were mock inoculated with a solution of 0.1% Tween 80. The experimental setup consisted of six treatments (five *M. brunneum* isolates, one control) with 12 replicate plants per treatment. A complete randomized design was used.

Four weeks after the inoculation of the spores, plants were artificially infested with eight CRF eggs per plant. The eggs were placed 1 cm below the surface on the root collar of each plant using a fine camel hairbrush, and eggs were covered by a thin layer of soil. The hatchling larvae were allowed to feed on the roots for 30 days. At this point, larval development was expected to be completed. The pupae were recovered as previously described. The mycosis of pupae was evaluated at 2-day intervals. Emerging adults were placed on a new plate to avoid cross-contamination. The difference between inserted eggs and recovered pupae was counted as *missing larvae*, assuming they died at the egg or larval stage. Total mortality (%) was calculated as
$$Total\;motality\;\%\;=\frac{\;\left(missing\;larvae\right)+\left(mycosed\;pupae\right)+\left(mycosed\;flies\right)}{introduced\;eggs}\times 100$$

Damage to the root collar of the plant was visually evaluated using a scale ranging from 0% (no damage) to 100% (whole collar root surface damaged) according to [40]. The amount of damage caused by each surviving pupae was estimated by the ratio damage (%) to the number of pupae recovered in each pot/root compartment.

### 2.4. Quantification of Rhizospheric and Endophytic Metarhizium Brunneum

To evaluate fungal colony forming units (CFUs) in the rhizosphere, three lateral roots with the soil still attached were placed in 50 mL Falcon tubes (Sarstedt AG & Co. KG, Nümbrecht, Germany) with 25 mL of 0.1% Tween 80. Tubes were vortexed for 10 s and inverted five times every 30 min for 3 h. After sedimentation for 20 s, 0.1 mL of a 1:10 dilution was plated on 9 cm Petri dishes with a semi-selective medium [41]. Petri dishes were then incubated at 23 °C and 65% RH for 21 days in darkness. The fungal colonies were counted every 3 days starting 10 days after plating until no new colonies appeared. Colonies of *M. brunneum* were identified by their morphology. This procedure was slightly modified in the split-root experiment (Section 2.5, Section 2.6 and Section 2.7): after adding Tween 80 to the samples, the tubes were vortexed for 10 s and placed on a shaker in horizontal position for 20 min at 250 rpm. Samples were sonicated for 30 s, briefly vortexed, left to sediment for 20 s, and then plated as described above. This shortened the processing time and the sonication step ensured better spore release from soil particles [42].

Endophytic colonization was measured with real-time quantitative PCR (qPCR) in a 2 cm segment of root collar from each plant (Section 2.3) or from each half of the split-root (Section 2.5, Section 2.6 and Section 2.7). Root segments were surface sterilized with 70% ethanol for 1 min and 2% sodium hypochlorite for 5 min and rinsed three times with sterile water for 30 s. Roots were frozen at −25 °C for 24 h, lyophilized in a freeze dryer (Martin Christ Freeze Dryers, Osterode am Harz, Germany) for 72 h, and milled with a mixer mill (Retsch MM 200) in a stainless-steel container with a 20 mm/32 g steel sphere (Retsch GmbH, Haan, Germany) for 30 s at maximum speed. DNA was extracted from 30 mg of root tissue using the cetyltrimethylammonium bromide (CTAB) buffer extraction method described previously [43]. DNA quality was verified on agarose (0.8%) gels. The CFX384™ Real-Time System with a C1000™ Thermal Cycler (BioRad, Hercules, CA, USA) was used for fungal DNA amplification and melting curve analysis. The primers used were specific for *Metarhizium* clade 1: Ma 1763 (CCAACTCCCAACCCCTGTGAAT) and Ma 2097 (AAAACCAGCCTCGCCGAT) [44]. Amplification was performed with 1:10 dilutions of the DNA extracts. The reaction mix contained the following: 5 µL of 2x qPCRBIO SyGreen Low-ROX (PCRBIOSYSTEMS), 0.2 µL of 10 µM of each primer, 1 µL of DNA template solution, and 3.6 µ of water to complete a final volume of 10 µL. Running conditions were: denaturation for 2 min at 95 °C, 40 cycles of a 5 s at 95 °C, 20 s at 66 °C, and a 10 s at 72 °C, with a final step at 72 °C for 5 min. Melting curves were obtained by increasing the temperature to 95 °C for 60 s and decreasing it to 55 °C for 60 s with a subsequent temperature increase from 55 °C to 95 °C by 0.5 °C per cycle with continuous fluorescence measurement. Absolute fungal DNA per g of plant tissue was measured by comparing threshold cycle (Ct) values against DNA standards starting with a concentration of 100 pg µL^−1^ and decreasing with a 1:3 dilution factor. The threshold cycle and standard curves were generated by the Bio-Rad CFX Maestro software. The identity of the amplicon was verified by comparing its size using gel electrophoresis. The presence of DNA in the root was evaluated for 12 replicate samples per isolate (Section 2.3) or 24 samples per treatment, 12 from each root compartment (Section 2.5, Section 2.6 and Section 2.7).

### 2.5. Direct and Systemic Effects of M. brunneum Isolate Gd12 on CRF Survival and Root Damage: Split-Root Setup and Bioassay

A split-root system was designed to differentiate between the direct effects caused by fungal infection and the indirect effects as a result of induced changes in plant metabolism. Gd12 was used, as this isolate had shown the strongest negative impact on CRF in the previous tests. To obtain seedlings with more than one tap root for the split setup, surface sterilized seeds were placed in Petri dishes with half strength Murashige and Skoog basal medium (Sigma-Aldrich, St. Louis, MO, USA), which contained sucrose and was adjusted to a pH of 6.8. The final concentration of agar was 10% (*w*/*v*). Petri dishes were then placed in a vertical position at 20 °C in darkness. Three days later, the seedling taproot was removed at the root base and the lower half of the Petri dish was covered with a light blocking fabric and left in a vertical position for five more days. Afterwards, seedlings were transplanted into a split-root system consisting of a pair of square pots (11 × 11 cm, 1.5 L) filled with non-sterile soil mix. The seedling was held in the center with a polypropylene cylinder (made from a 3 mL pipette tip; 13 mm Ø × 25 mm), and seedlings were inoculated by drenching the roots with 1 mL of either spore suspension or 0.1% Tween 80 as a control. The experimental setup consisted of five treatments with 12 replicate plants per treatment:(1)Control—one compartment mock-inoculated with 0.1% Tween 80;(2)Fungus—compartment A: *M. brunneum* (Mb-L), B: 0.1% Tween 80 (Mb-S);(3)Herbivore—compartment A: CRF (Dr-L), B: 0.1% Tween 80 (Dr-S);(4)Fungus/herbivore (local)—compartment A: *M. brunneum* + CRF (Mb-L/Dr-L), B: 0.1% Tween 80 (Mb-S/Dr-S);(5)Fungus/herbivore (systemic)—compartment A: CRF (Dr-L/Mb-S), B: *M. brunneum* (Mb-L/Dr-S).

In the treatment codes above “L” and “S” refer to the local and systemic roots, respectively. The position of the plants in the growth chamber was completely randomized. The plants were kept under the same conditions as described above. Infestation with CRF eggs, pupal recovery, and damage assessment were carried out as described in Section 2.3; however, only four eggs per root compartment were used in this experiment.

### 2.6. Direct and Systemic Plant Responses to Fungus and Herbivore: Gene Expression

Gene expression was analyzed in root tissues obtained from the split-root setup described in Section 2.5. Plants were harvested 7 days after egg inoculation when larvae had fed for about 24 h. Roots were washed and a 2 cm collar root segment from each root compartment was sliced, snap-frozen in liquid nitrogen, lyophilized, and milled as described in Section 2.4. 

Total RNA was extracted with TRIzol (Invitrogen, Carlsbad, CA, USA) from 20 µg of lyophilized ground plant tissue following the manufacturer’s instructions. The integrity of RNA was evaluated by denaturing gel electrophoresis. Concentration and purity were assessed by checking the absorbance ratios of OD260/OD230 and OD260/OD280 using a microplate spectrophotometer (Epoch, Bio-Tek (Agilent), Santa Clara, CA, USA). The first strand of cDNA was synthesized from 1 µg of total RNA using Fast Gene^®^ Scriptase II (Nippon Genetics Europe, Düren, Germany) using a mix (1:0.5 by volume) of oligo dT and random hexamers following the manufacturer’s instructions. 

Most of the primers used were published in previous studies (Appendix A). Primers for the *BABG*, *BnMyr4*, *Myr2.*Bn1, and *DTCMT*.a genes were designed with Primer3 (v.4.1.0) [45] using *B. napus* specific gene sequences from Genbank (https://www.ncbi.nlm.nih.gov/genbank/) and Plaza 4.0. (https://bioinformatics.psb.ugent.be/plaza/versions/plaza_v4_dicots/, accessed on 8 August 2020) databases. 

Gene transcripts were measured by qPCR as described in Section 2.4, with the following specific conditions. The reaction mixture contained the equivalent of 5 ng total RNA, 5 µL of 2× qPCRBIO SyGreen Low-ROX (PCR biosystems^TM^, London, UK), 0.2 µL of 10 µM of each primer, 1 µL of DNA template solution, and 3.6 µL of PCR-grade water to complete a total 10 µL final volume. The following temperature program was run: 95 °C for 2 min, 40 cycles of 95 °C for 10 s, and 60 °C for 30 s. Amplicon specificity was controlled by melting curve analysis as previously described. The relative expression of each gene was calculated using the 2^−^^ΔΔCT^ method, with correction for primer efficiency [46], normalized to the endogenous reference gene *ACTIN*, and subsequently normalized to those in the control plants. The selected genes are involved in defense responses (See Appendix A for gene description). As markers for phytohormonal pathways we used *ABA2* for abscisic acid (ABA); *ACO* and *ERF2* for ethylene (ET); *PAL* and *PR1* for SA; and *AOS*, *MYC2*, *PDF 1.2*, and *TPI for* JA. As markers for glucosinolate (GSL) metabolism, we used *CYP83A1* and *CYP79B2* for indol and aliphatic GSL, respectively, *GTR1A2* for GSL transport; BABG, *Myr2.Bn*, and *BnMyr4* for GSL degradation; and *BnDTCMT.a* for the phytoalexin brassinin.

### 2.7. Direct and Systemic Plant Responses to Fungus and Herbivore: Phytohormone Analysis

Phytohormones were analyzed in root tissues obtained from the split-root setup described in Section 2.5. The extraction was performed following a modified method of Müller and Munne-Bosch (2011) [47]. Briefly, 40 mg of freeze dried and ground root tissue was suspended in 0.5 mL cold extraction solution consisting of methanol/isopropanol (20:80, *v*/*v*) with 0.1% formic acid (*v*/*v*), and placed in an ultrasonic bath at 4 °C for 10 min. Afterwards, samples were shaken at 280 rpm at 4 °C for 2 h and centrifuged at 13,000 rpm at room temperature for 10 min. The supernatant was further cleaned by centrifugation twice for 10 min at 13,000 rpm. Then, 200 µL of supernatant was transferred into a HPLC amber glass vial and analyzed instantly. The following chemicals were used as authentic standards: jasmonic acid (Cayman Chemical, Ann Arbor, MI, USA), salicylic acid (Sigma-Aldrich, Steinheim, Germany), salicylic acid glucoside (synthesized in the Division of Molecular Phytopathology and Mycotoxin Research, University of Göttingen, Gottingen, Germany), and abscisic acid (Sigma-Aldrich, Steinheim, Germany). HPLC-MS was used for the quantitative analysis of phytohormones. The system consisted of 1290 Infinity II UHPLC (Agilent Technologies, Waldbronn, Germany) equipped with a Zorbax Eclipse Plus C18 column (1.8 µm; 50 × 2.1 mm; Agilent Technologies; column temperature: 40 °C). The HPLC system was coupled to a 6460 triple quadrupole mass spectrometer (Agilent Technologies, Waldbronn, Germany) with an electrospray ion source (capillary voltage, 4 kV; nebulizer pressure, 60 psi; nitrogen flow, 13 L min^−1^; nitrogen temperature, 350 °C). The separation was achieved through a binary gradient elution program at a flow rate of 0.4 mL min^−1^. The mobile phase A was water/formic acid (99.9:0.1, *v*/*v*), and B was methanol/formic acid (99.9:0.1, *v*/*v*). The gradient program was as follows: isocratic B at 5% for 0.2 min; 0.2 to 6 min, a linear increase from 5% to 75% B; 6 to 6.50 min, from 75% to 98% B; 6.50 to 9 min, isocratic B at 98%; 9 to 9.50 min, return to 5% B; 9.5 to 13 min, isocratic B at 5% for re-equilibration. The injection volume was 5 µL. Phytohormones were quantified in multiple reaction monitoring mode (MRM). The acquisition details are listed in Appendix A. The calibration curve included 12 points from 0.48 to 1000 μg L^−1^. Blanks and quality control standards were analyzed regularly. Limit of quantification (*LOQ*) and limit of detection (*LOD*) values were determined based on the standard deviation of the blank [48] with the following formulas:$$LOD=3.9\times \frac{\mathrm{Sy}\;}{b}$$
LOQ=3.3×LOD 
where Sy = standard deviation of the blank and b = slope of the calibration curve.

### 2.8. Data Analysis

Data exploration and statistical analyses were performed with R 4.0.3 [49]. We used generalized linear models (GLM, library MASS [50]) with a binomial family distribution to evaluate adult emergence, pupae recovered per plant, and CRF total mortality (including larvae and pupae with mycosis as well as adults that developed mycosis after emergence). The percentage of damage that larvae inflicted to the root collar, and the ratio of damage per pupa were analyzed with a beta regression model for percentages (library betareg [51]). For DNA and CFU values we used linear models with a quasi-Poisson family distribution due to the over dispersion of the data. Pupal weight was analyzed by one-way ANOVA after confirming normal distribution and homoscedasticity of residuals. Results from pupal survival were corrected using Abbott’s formula [40] which standardizes results based on control mortality. Gene expression data and phytohormone concentrations were analyzed by linear models, using compartment as a factor. All, except *ACO* transcript data, were logarithmically transformed to meet the assumptions of normality. When a model was significant, we used Tukey’s honest significance test (Library multcomp [52]) for post-hoc analysis.

## 3. Results

### 3.1. Susceptibility of CRF to Different M. brunneum Isolates in Substrate

All of the *M. brunneum* isolates tested caused significant mortality in CRF larvae (L_3_). The isolates Gc1I and Cb15III significantly reduced the percentage of emerged adults when larvae were exposed to conidia in non-sterile soil mix when compared with control treatment (Figure 1a). If the substrate was not considered in the model, Gc1I and Cb15III were significantly different from the control (GLM binomial model, *p* = 0.002 and *p* = 0.025 respectively). However, a significant effect was also found for the interaction fungal isolate × substrate (*p* = 0.049). In non-sterile soil substrate, we observed reduced adult eclosion (Chi square test, Holm *p*-Value adjustment method, *d f* = 5, χ^2^ = 19.12, *p* = 0.004).

When pupae and adults with symptoms of mycosis were included in the analysis, all isolates caused significantly higher mortality in CRF individuals compared to the control (Figure 1b). This was true for both soil and sand substrates (Appendix A). Abbott’s corrected mortality was highest in Gd12 and lowest in Cb15III. With exception of the isolate Gc1I, significantly higher total mortality of CRF individuals was found in sterile sand compared to non-sterile soil mix (*p* = 0.005). 

### 3.2. Rhizosphere Competence, Endophytism, and Plant Protection of M. brunneum Isolates

In this pot experiment, all five *M. brunneum* isolates were detected in the plant rhizosphere and in root tissue. Fungal CFUs varied from 84 spores g^−1^ of soil for Gd12 to 1963 spores g^−1^ for Gc1I, the latter being statistically different from the other isolates (Figure 2a; GLM, *p* < 0.001, Tukey HDS). Fungal DNA per gram of root collar tissue varied from 0.32 ng in CC5 to 2.07 ng in Gd12, the latter having significantly higher fungal DNA content than in isolates Cb17B and CC5 (Figure 2b; GLM, *p* < 0.001, Tukey HDS). In the control treatment, we found no CFUs in the rhizospheric soil and no fungal DNA in the root tissue.

*Metarhizium brunneum* isolates Cb15III, Gc1I and Gd12 significantly increased CRF total mortality when compared to the control (Figure 2c, Appendix A). Abbott’s corrected mortality was 20.0% for Gd12, 17.5% for Cb15III, and 15.0% for Gc1I. Additionally, all fungal isolates significantly reduced root damage compared to the control (Figure 2d, Appendix A). Furthermore, we found a negative correlation between the amount of fungal DNA in the root collar tissue and the number of pupae recovered per plant at the end of the experiment (Figure 2f, Pearson’s test, *r* = −0.42; *p* = 0.0019). We also detected significantly lower root damage per surviving pupae in isolates Cb17B, Gc1I, and Gd12 (Figure 2e). 

### 3.3. Direct and Systemic Effects of M. brunneum Isolate Gd12 on CRF Survival and Root Damage: Bioassay 

Plants successfully developed two similar root systems in our split-root setup. Root drenching of the seedling roots resulted in the colonization of the plant rhizosphere and root collar tissue. The fungus was only detected in the rhizosphere and roots of the compartments that were inoculated with *M. brunneum* (Figure 3a,b). No fungal CFUs in the rhizospheric soil or fungal DNA in the root tissue were detected in the untreated compartment of the split-root setup. The fungus was also absent in the controls.

*Metarhizium brunneum* (Gd12) caused a significant reduction in the number of CRF pupae when present in the same soil compartment as the larvae. The number of pupae recovered in roots of the systemic treatment was not significantly different from the control (Figure 3c, GLM binomial family, *p* (local) = 0.024; *p* (systemic) = 0.176). Abbot’s corrected mortality was 32% in the local treatment and 16% in the systemic treatment. However, we observed a significant reduction in root damage (%) in both the local and systemic treatment when compared to the control (Figure 3d, beta regression model, *p* (local) < 0.001; *p* (systemic) = 0.03). Pupal weights did not differ significantly between treatments (control: 15.8 mg; local treatment: 16.2 mg, systemic treatment: 15.4 mg; *F* (2, 33) = 0.51, *p* = 0.603).

### 3.4. Direct and Systemic Plant Responses to CRF and M. brunneum: Gene Expression and Phytohormone Analysis

Local feeding by CRF larvae (Dr-L) led to an increased JA concentration (χ^2^ = 6.07, *p* < 0.001, Figure 4a) and an upregulation of the JA biosynthesis gene *AOS* (χ^2^ = 5.92, *p* < 0.001, Figure 5a) in all root compartments with CRF. Genes in JA downstream signaling were also upregulated as shown by higher expression of *MYC2* (χ^2^ = 8.38, *p* < 0.001; Figure 5b), *TPI* (χ^2^ = 10.06, *p* < 0.001; Figure 5c) and *PDF 1.2* (χ^2^ = 4.16, *p* < 0.001; Figure 5d). Larval feeding also induced the salicylic acid (SA) pathway but only at the gene expression level. While no increased concentrations of SA metabolites were observed (Figure 4c,d), there was an enhanced transcription of *PAL* (χ^2^ = 4.44, *p* < 0.001; Figure 5g). Enhanced expression was also measured for *PR1* (χ^2^ = 5.42, *p* < 0.001 Figure 5h). Abscisic acid (ABA) concentrations were higher in the compartment with larval feeding than in the control plants (χ^2^ = 2.30, *p =* 0.026 Figure 4b) but *ABA2* was not upregulated (Appendix A). The ACO ethylene synthesis gene also showed higher levels of transcripts in response to insect feeding (χ^2^ = 2.46, *p =* 0.018 Figure 5e) but not the downstream signaling gene *ERF.2* (Figure 5f).

CRF feeding also activated the glucosinolate (GLS) defense system. The indole GLS synthesis gene *Cyp79B2* (χ^2^ = 12.35, *p* < 0.001 Figure 5j) and the phytoalexin gene *DTCMT* (χ^2^ = 10.69, *p* < 0.001 Figure 5k) showed enhanced expression when CRF larvae were present in the root compartment. *GTR1A2* was also upregulated (χ^2^ = 4.11, *p* < 0.001 Figure 5i). Nevertheless, the expression of *CYP83A1*, involved in the synthesis of aliphatic GLS was similar to control plants (Appendix A). Of the three myrosinase biosynthesis genes evaluated, only *Myr2Bn* (χ^2^ = 2.56, *p =* 0.03 Figure 5l) was slightly upregulated by herbivory (other myrosinases, Appendix A). 

In general, biochemical responses of roots to CRF were limited to the root compartment with herbivory (Dr-L). We only detected a higher JA concentration in the systemic compartment (Dr-S) when compared with the control. (Dr-S; χ^2^ = 6.47, *p* < 0.001, Figure 4a). However, there was no change in the gene expression of any of the JA-associated genes (Figure 5a–d). Likewise, we did not detect a change in SA and ABA concentration (Figure 4b–d) or gene expression (Mb-S, Figure 5g,h and Appendix A) in the systemic compartment. The ethylene response factor *ERF.2* was downregulated (Dr-S, χ^2^ = −4.53, *p* < 0.001, Figure 5f). Glucosinolate biosynthesis was not activated (Dr-S, Figure 5j), but there was upregulation of the GSL transporter gene *GTR1A2* (Dr-S, χ^2^ = 2.70, *p* ≤ 0.001, Figure 5i). None of the genes involved in myrosinase synthesis were affected in the systemic compartment (Figure 5l and Appendix A). 

Inoculation with *M. brunneum* alone (Mb-L) did not increase phytohormone concentrations (Figure 4) or the expression of genes involved in phytohormone biosynthesis or signal transduction (Mb-L, Figure 5), when compared to the control treatment. The expression of *GTR1A2* was upregulated (χ^2^ = 3.23, *p* = 0.002, Figure 5i) but the expression of myrosinase synthesis genes did not differ from the control. On the other hand, systemic roots (Mb-S) showed higher JA concentrations (χ^2^ = 4.69, *p* < 0.001, Figure 4a) and a lower gene expression of *ERF2* (χ^2^ = −3.64, *p* < 0.001, Figure 5f). 

*Metarhizium brunneum* inoculation modified the jasmonate response induced by CRF both locally and systemically. Concentrations of JA increased 1.8-fold when the fungus colonized the same roots as the larvae and 1.6-fold when the fungus was in the systemic compartment (Mb-L/Dr-L and Dr-L/Mb-S, respectively, Figure 4a). At the gene expression level, the presence of the fungus in the adjacent compartment of the herbivore induced a higher expression of *PDF 1.2* (Dr-L/Mb-S, Figure 5d). *Metarhizium brunneum* did not modulate the expression of *GTR1A2*, *Cyp79B2*, *Cyp83A1*, or *DTCMT* in response to larval feeding (Mb-L/Dr-L and Dr-L/Mb-S). However, *Myr2Bn* was upregulated when *M. brunneum* was in the systemic compartment (Dr-L/Mb-S).

## 4. Discussion

Fungal endophytes and herbivores often associate with the root system of the same host plant. Therefore, the differentiation between the direct and plant-mediated effects of endophytic entomopathogens on herbivores is often not straightforward. To address this question, we first assessed the interactions between five *M. brunneum* isolates and CRF in the absence and presence of the host plant *B. napus*. The results showed that several fungal isolates were highly pathogenic in the soil substrate without plants and, after colonizing the roots, were capable of decreasing insect survival and plant damage. We then used a split-root setup to study the systemic plant-mediated effects of the isolate *M. brunneum* Gd12, and the results suggested that local and systemic plant defense responses against CRF play a role and can be primed.

Previous studies have shown that CRF is highly susceptible to different *Metarhizium* isolates in in vitro experiments. These bioassays were performed in the absence of soil [30,32,34] or in sterile substrates [31,33]. However, soil substrates may harbor a wide range of microorganisms that influence entomopathogenic fungi and their interactions with insects. As a result, the virulence of entomopathogens may be overestimated in sterile substrates [53,54,55]. Experiments with non-sterile soil were therefore carried out to confirm the pathogenicity of *M. brunneum* under more realistic conditions and to study isolate-specific effects. The bioassays showed significant reductions in adult emergence when infected with the isolates Cb15III and GC1I and there was high mortality in adult flies for all isolates. The infected flies died shortly after eclosion and developed mycosis. So, overall, the tested isolates performed well in a non-sterile environment. It should also be noted that mycosis after adult fly eclosion has not previously been reported. From a biological control perspective, this aspect could be advantageous, since targeting juvenile CRF with soil application of *M. brunneum* may lead to increased infection rates in adult fly populations. The transmission of conidia between adult flies has been observed before and it is known that adult flies are susceptible to dry spores of *M. anisopliae* [56].

In the experiment where *M. brunneum* was given the opportunity to colonize *B. napus* roots, three isolates were identified that significantly reduced pupal survival. All isolates though reduced the root collar area damaged by CRF larvae and were able to establish in the plant rhizosphere. Decreased herbivore damage was also observed under field conditions when cabbage plants were inoculated with *M. anisopliae* [29]. We expected lower pupal numbers in treatments where the fungal isolates showed higher colonization of the rhizosphere, as this may translate into more fungal spores being available to infect the insects. No negative correlation between CFUs and recovered pupae was apparent. However, this correlation was found for fungal DNA in the root tissue and the number of pupae recovered at the end of the experiment (see Figure 2f), confirming our expectation that endophytism may contribute to negatively affecting larvae. It is conceivable that more fungal biomass leads to stronger root defense priming. Nevertheless, this result must be taken with caution, as the negative correlation is driven mainly by two isolates. 

Intraspecific variation in fungal performance is a common feature and fungal isolates may differ in their virulence to insects [30,32,57], in rhizosphere competence [58], rhizoplane colonization [58,59], and endophytism [7,58]. Therefore, the selection of several isolates is necessary to understand the variability between isolates of a single fungal species. In this study, *M. brunneum* Gd12 was the most successful fungal isolate that caused high in vitro and in planta mortality in CRF. Interestingly, it was also the isolate with the lowest CFUs in the soil and the highest level of endophytic colonization. Hence, Gd12 was used in the split-root experiment to test whether the adverse effects of *M. brunneum* against CRF larvae were enforced by changes in plant metabolism. 

Our observations confirmed that in the split-root setup, the fungus did not grow systemically in the roots of the adjacent compartment. Therefore, we assumed that the performance of CRF on systemic treatment was not the result of any direct interactions with the fungus. As in the previous experiment, *M. brunneum* Gd12 reduced both the number of pupae and the extent of plant damage by larvae in the local compartment. Similar effects were also found for the systemic compartment, although only the decrease in plant damage was statistically significant. 

The negative effects on CRF that fed on roots in the systemic compartment (Dr-L/Mb-S) may be explained by defensive root responses as a result of fungal colonization and herbivore feeding. In general, the roots showed significantly increased JA concentrations in response to CRF, while the effects of *M. brunneum* on JA were considerably smaller. However, the highest concentrations of JA were measured in plants affected by both organisms, and this was irrespective of whether the fungus and herbivore were in the same (local) or different (systemic) compartments of the root system. The increased JA levels were accompanied by the upregulation of the myrosinase synthesis gene *Myr2Bn,* and the plant defensin gene *PDF1.2*, with higher expression in response to herbivory only when the fungus was in the systemic compartment (Dr-L/Mb-S). Interestingly, a priming response involving myrosinase enzymes was recently reported, where *M. brunneum* inoculation induced higher myrosinase activity in cauliflower plants attacked by *P. xylostella* [8]. Moreover, the upregulation of *PDF1.2* was also observed in leaves of oilseed rape plants inoculated with another fungal endophyte, *Trichoderma harziarum*, in response to *Sclerotinia sclerotium* [60]. Therefore, our results suggest that the endophytism of *M. brunneum* primes the root for a stronger induction of JA upon herbivory. 

Apart from JA, the phytohormones ABA and SA may be involved in plant responses to CRF [61]. Abscisic acid signaling regulates responses to herbivory by co-activating the MYC branch of the JA pathway [62,63]. However, in the present study, no clear pattern of ABA involvement was found (Figure 4b). Likewise, fungus and herbivore had no effects on the SA pathway as neither SA nor its glucosylated form (SA glu) were increased in the different treatments, when compared to control roots (Figure 4c,d). The SA pathway is important in the plant defense response to biotrophic pathogens [62] and also plays a role in modulating systemic responses to sap-sucking or cell content-feeding insects such as aphids and white flies [64,65] and also chewing herbivores [61,66]. The observed upregulation of PAL and PR1 genes in our study may imply an involvement of SA signaling since PAL is at the base of the phenylpropanoid pathway that leads to SA biosynthesis. However, the phenylpropanoid pathway is also responsible for the biosynthesis of defense-related compounds such as lignin, coumarins, anthocyanins, and flavonoids [67]. The pathogenesis-related protein PR1, a marker for SA downstream signaling, was strongly induced in all roots that were damaged by herbivory, irrespective of fungal presence in local or systemic roots. The results presented here confirm those of Karssemeijer et al., (2020) [61] who observed the increased expression of PAL in B. oleracea after 6 h and PR1 upregulation as a contributor in the Partial Least Squares Discriminant Analysis model of plant response to CRF herbivory at 24 h. However, PR-1 induction has also been reported as a response to *P. xylostella* herbivory in wild *B. oleracea* [66]. Pathogenesis-related genes are also induced by cys-nematodes [68]. Their activation in response to CRF larvae and cys-nematodes could have similar purposes; both organisms cause damage to the root tissues. Therefore, SA activation could also provide protection against pathogen infection. It would be interesting to explore the role of other components of the phenylpropanoid pathway in plant defense against specialized Brassica herbivores, such as CRF. We observed a red–purple coloration of the root in the feeding zone in plants harvested for biochemical analysis (~24 h after initial feeding), suggesting the presence of anthocyanins. Furthermore, other Brassica specialists such as *Pieris brassicae* and *Phyllotreta nemorum* upregulated phenylalanine, flavonoids, and phenolic acids [69]. Furthermore, lack of induction of kaempferol-3,7-dirhamnoside levels after MYB75 overexpression, led to a loss of resistance of A. thaliana plants to *P. rapae* feeding [70] and QTL resistance of canola to *Ceutorhynchus obstrictus* (cabbage seed pod weevil) correlated with a peak of the flavonoid kaempferol 3-O-sinapoylsophoroside 7-O-glucoside [71].

Glucosinolates are characteristic defense compounds in Brassicaceae that can be induced by herbivory via the JA pathway [72,73,74,75]. The data presented here confirmed the upregulation of GSL marker genes in response to CRF [75,76]. Larval damage to the roots of *B. napus* strongly increased the transcription of the indole-GSL gene *CYP79B2* and of *DCT-MT.* The latter gene encodes a dithiocarbamate S-methyltransferase, which catalyzes the final step in brassinin biosynthesis. DCT-MT induction in response to CRF has not been reported previously though. The phytoalexin brassinin contains a dithiocarbamate group that has insecticidal properties [77] and may thus play a role in defense against CRF larvae. Local induction of the myrosinase gene *Myr2Bn* was also found predominantly in roots with herbivore presence, although this was only statistically significant for the Dr-L/Mb-S treatment. Although induced by herbivory, GLS does not appear to greatly affect CRF development or feeding behavior. This is probably due to the microbiome in the larval gut that is capable of detoxifying GLS-derived isothiocyanates [73,78]. However, for the attacked plant, activation of the GSL defense system in the wounded roots could be a strategy to prevent secondary infection by microbial pathogens [75,79]. 

## 5. Conclusions

This study showed that *M. brunneum* infects various life stages of the cabbage root fly and readily colonizes available plant roots. The capability of the different *M. brunneum* isolates to grow inside root tissue as an endophyte correlated positively with their impact on the herbivore, which suggests a supporting role for the plant in this interaction. By using a split-root setup, we demonstrated for the first time that endophytic colonization by an entomopathogenic fungus can prime plant defense responses against a root feeding herbivore, thus resembling similar interactions with other beneficial, root-associated microbes. The results further suggest an involvement of the JA pathway in the priming response against CRF, although the exact nature of the defense needs further exploration. Future studies using a split-root design and -omics approaches should help to find other genes and metabolites that take part in this plant–fungal–insect interaction. Additionally, it would be worthwhile to examine whether the priming response protects the plant against secondary soil-borne infections. In summary, we conclude that EPF as a biological control agent of the rhizosphere and as an endophyte can benefit plant health by multiple mechanisms. 

## Figures and Tables

**Figure 1 jof-08-00969-f001:**
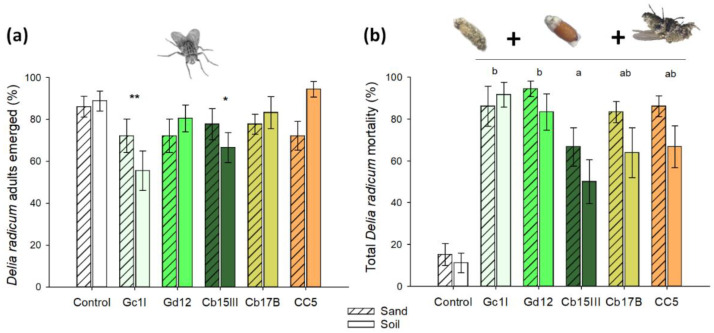
Susceptibility of *Delia radicum* (CRF) to isolates of *M. brunneum* in two different substrates (sterile sand = light grey bars; non-sterile soil substrate = dark grey bars). (**a**) Percentage of emerged CRF adults and (**b**) total mortality (larvae, pupae, and adult flies). The insects were exposed as larvae (L3) to substrate treated with fungal spores. Asterisks denote a significant difference from the control according to GLM, binomial distribution (*p* < 0.05) (Significance: ** *p* ≤ 0.01; * *p* ≤ 0.05). Mortality caused by the isolates with different letters in (**b**) differ significantly from each other (Tukey HSD test, *p* < 0.05). Data represent means ± SE; *n* = 12.

**Figure 2 jof-08-00969-f002:**
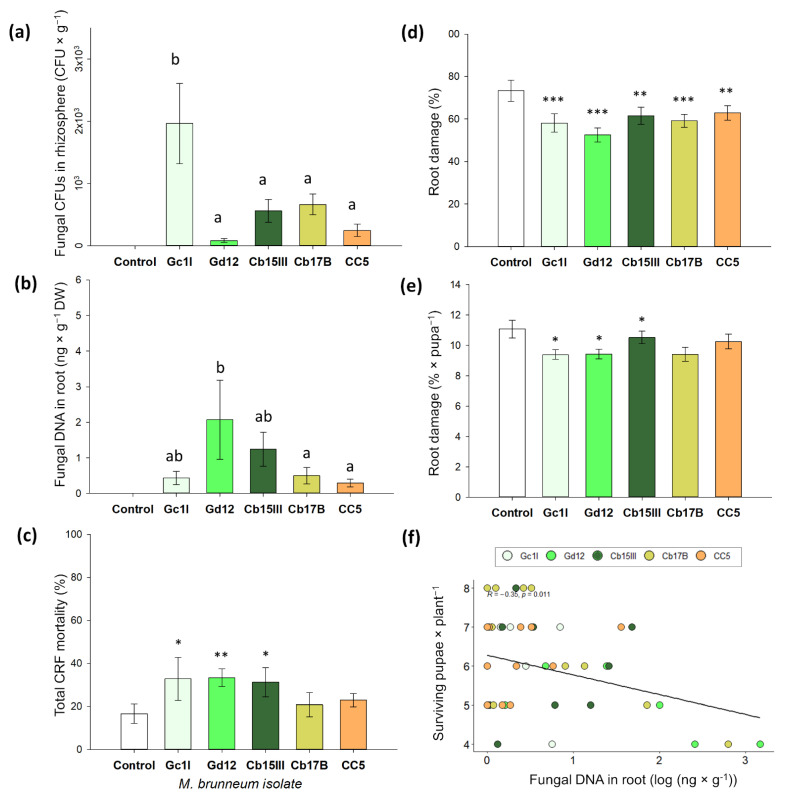
Colonization of rhizosphere and plant roots by isolates of *M. brunneum* and their effect on CRF survival and root damage. (**a**) Fungal colonization of rhizospheric soil (CFU g^−1^); (**b**) fungal DNA in root collar tissue (ng × g^−1^ DW); (**c**) total CRF mortality; (**d**) root collar surface damage (%); (**e**) root damage caused per pupa recovered; (**f**) Pearson’s correlation between *M. brunneum* DNA in root tissue and number of surviving pupae of CRF. Eight eggs of CRF were placed on the root collar of 5-week-old *Brassica napus* plants that had been inoculated with different *M. brunneum* isolates by root drenching at transplanting time (1 week). (**a**) Differences between isolates according to GLM with a quasi-Poisson family distribution (*p* < 0.05). Isolates with different letters differ significantly from each other (Tukey HSD test, *p* < 0.05). The asterisks above the columns indicate the significant difference from the control according to (**c**) GLM, binomial distribution (*p* < 0.05) and (**d**,**e**) percentage Beta-regression analysis (*p* < 0.05) (Significance: *** *p* ≤ 0.001; ** *p* ≤ 0.01; * *p* ≤ 0.05). Data represent means ± SE; *n* = 12.

**Figure 3 jof-08-00969-f003:**
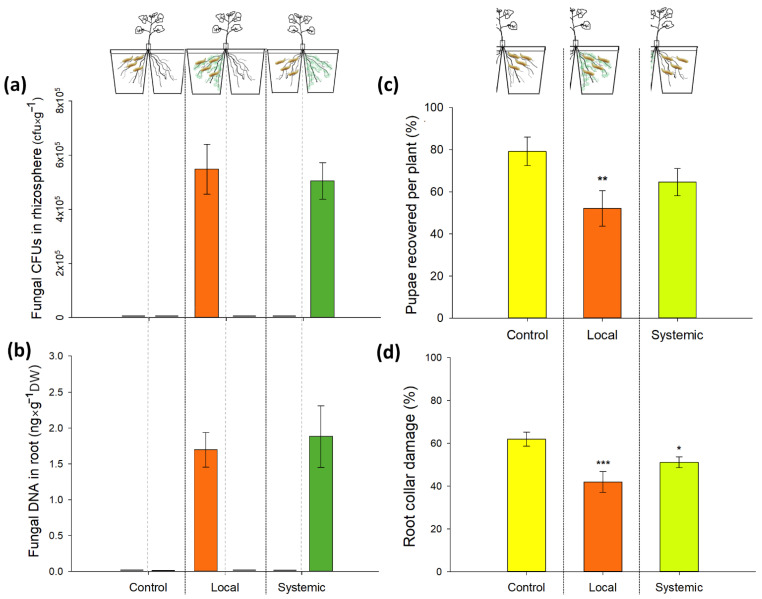
Local colonization of rhizosphere and plant roots by *M. brunneum* Gd12 and local and systemic effects on CRF survival and root damage. (**a**) Fungal colonization of rhizospheric soil (CFU g^−1^); (**b**) fungal DNA in root collar tissue (ng × g^−1^ DW); (**c**) CRF pupae recovered; (**d**) root collar surface damage (%). Four eggs of CRF were placed on the root collar of 6-week-old plants in a split-root setup that had been inoculated with *M. brunneum* Gd12 either in the local or systemic compartment by root drenching at transplanting time (10 days after sowing). Asterisks above bars indicate significant differences from the control according to binomial GLM (**c**) or according to percentage Beta-regression analysis (**d**), (*p* < 0.05), (Significance: *** *p* ≤ 0.001; ** *p* ≤ 0.01; * *p* ≤ 0.05). Data represent means ± SE; *n* = 12.

**Figure 4 jof-08-00969-f004:**
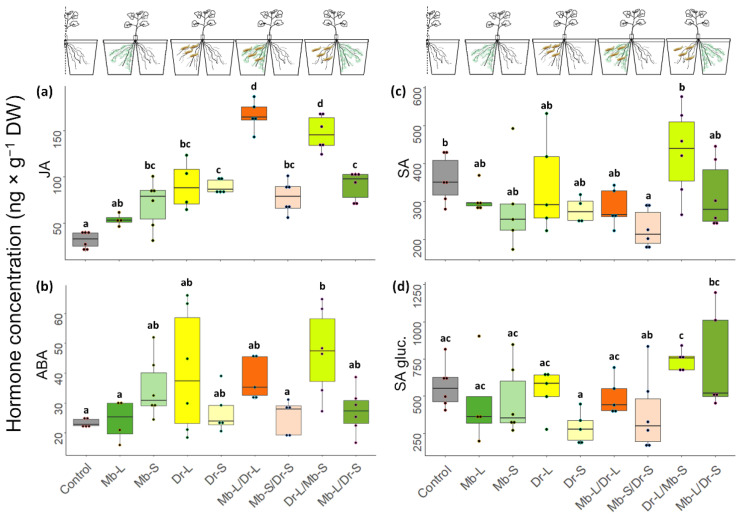
Phytohormone concentration of (**a**) jasmonic acid (JA), (**b**) abscisic acid (ABA), (**c**) salicylic acid (SA) and (**d**) salicylic acid glucoside (SA glu) in the tap roots of *B. napus* in a split-root setup. Local and systemic roots received the following treatments: mock inoculation with Tween 0.01% (control), *Metarhizium brunneum* in the local (Mb-L) or systemic (Mb-S) roots, CRF egg infestation in the local (Dr-L) or systemic (Dr-S) roots, both treatments in same roots (Mb-L/Dr-L), or different treatments on each split-root system (Dr-L/Mb-S; Mb-L/Dr-S). Plants were inoculated with *M. brunneum* at transplanting (10 d) and were harvested 7 days after egg infestation, 5 weeks after Mb inoculation. Different letters indicate statistically significant differences based on a linear model, and Tukey’s post-hoc test. Boxplots show the distribution of the data, where the lower, middle, and upper lines represent the first quartile, the median, and third quartile, respectively. Data points represent independent biological replicates (*n* ≤ 6).

**Figure 5 jof-08-00969-f005:**
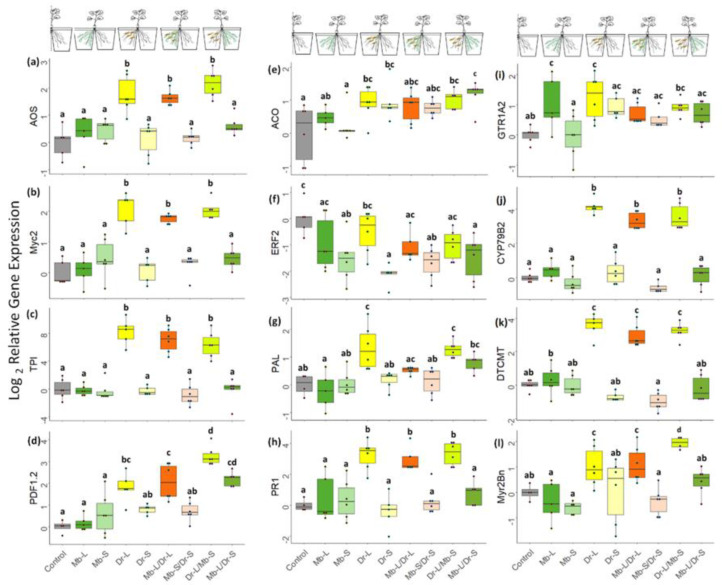
Expression of genes involved in defense signaling in taproots of *B. napus* in response to CRF herbivory or/and *M. brunneum* inoculation. Left panel shows genes related to jasmonic acid signaling (**a**–**d**). Central panel depicts genes involved in ethylene (**e**,**f**) and salicylic acid (**g**,**h**) signaling. The right panel shows genes involved in glucosinolate-related defense (**i**–**l**). The plants grew in a split-root setup in which each compartment had *M. brunneum* inoculation (Mb) in the local (L) or adjacent (S) compartment, CRF egg infestation (Dr) in the local (L) or adjacent (S) compartment, both treatments in the same compartment (Mb-L/Dr-L), or each in adjacent compartments of the same plant (Dr-L/Mb-S; Mb-L/Dr-S). Eggs were added 4 weeks after Mb inoculation. Plants were harvested 7 days after egg infestation. The letters represent statistically significant differences in the expression of the control treatment (Tukey HSD, *p* < 0.05). Boxplots show the distribution of the data, where the lower, middle, and upper lines represent the first quartile, the median, and third quartile, respectively. Data points represent independent biological replicates (*n* ≤ 6).

## Data Availability

Data will be available upon request.

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
