# Peer review of "Root Colonization by Fungal Entomopathogen Systemically Primes Belowground Plant Defense against Cabbage Root Fly"

_jof, 2022, doi:10.3390/jof8090969_

Round 1
Reviewer 1 Report
The manuscript titled “Root colonization by fungal entomopathogen systemically primes belowground plant defense against cabbage root fly” contains some important information.
The abstract needs revision. The authors should add some data in the abstract. The key components of the methodology should also be mentioned.
Keywords correspond to the aim.
The introduction is specific and focused on. However, the objectives need revision.
Line 85-90: These are redundant.
The methodologies have been described in more detailed. These should be squeezed. In this section chapter should be replaced with sections e.g. section 2.5, 2.6, 2.7 etc.
Results are quite interesting and analysis is strong; well written and explained.
Discussion confirmed results very well and is a logical explanation of results.
Conclusion needs revision and should be strengthened with data. The authors should give some recommendation on the basis of their findings.
Numerous stylistic errors have also been spotted.
References are adequate and need to be crosschecked. Some of the journal abbreviations are not correct.
In some cases, the authors used first form of speech. Usually results are written in the third form of speech with passive voice e.g.
Line # 22: --- We conclude that root----
Line # 80: aim of our study was to compare---------
Line # 88: Our results suggest that--------
Line # 115: To obtain larvae, we placed 100 ---------
Line # 122: We evaluated the pathogenicity of M. brunneum towards----------
Line # 128: We evaluated---
Line # 142: We evaluated the rhizosphere competence-------
Line # 214: We used M. brunneum Gd12 as this isolate-----------
Line # 268: we used ABA2 for abscisic---------
Line # 328: we observed reduced adult eclosion
Line # 354: we found no CFUs in the----
Line # 380: systems in our split-root setup.
Line # 383: We neither measured fungal----
Line # 384: we detect fungal
Line # 455: We only detected higher-------
Line # 458: Likewise, we did not detect a change-----------
Line # 485: we first assessed the interactions---------
Line # 486: Our results showed that several
Line # 488: We then----
Line # 499: Our bioassays showed significant
Line # 500: we found high mortality----
Line # 509: In our experiment,
Line # 513: We expected-----------
Line # 518: confirming our expectation 518---------
Line # 524: In our study, M. brunneum
Line # 529: Our observations confirmed----
Line # 530: Therefore, we assumed that the
Line # 532: As in our previous experiment,
Line # 550: we detect fungal
Line # 554: However, in our study, we found no clear
Line # 567: Our results confirm---------
Line # 576: We observed
Line # 586: Our data confirmed----
Line # 601: Our study shows
Line # 605: we demonstrated for the first time
Line # 610: we conclude that EPF as a biological
In some cases, present tense was used.
There is no consistency in writing scientific names. Usually, the genus name is written in full at its first mention and subsequently abbreviated. But this was not followed in the present manuscript.
Reviewer 2 Report
This manuscript presents very interesting data regarding a complicated fungus-herbivore-plant interaction and reports how the entomopathogenic endophytic fungus M. brunneum induces plant defense and reduces herbivore's (cabbage root fly's) damage by root colonization. The effects of fungal root colonization on the plant defense and the herbivore damage are well shown with robust data from well-done split-root experiments, in which five fungal isolates were tested in the presence or absence of herbivore. I enjoy reading the manuscript and believe that the results and conclusion will benefit our colleagues in the society. However, a revision is still needed for improved clarity and conciseness. I suggest the authors to consider the following points when they revise their manuscript.
1. It is unnecessary to frequently mention the scientific names Delia radicum and Metarhizium brunneum in the text. The use of the abbreviations CRF and EEF defined in the first presence of each would increase readability (e.g., CRF susceptibility (eggs, larvae, pupae, adults, ...) or EEF isolates) throughout the text without any confusion. This is becuase the experimental system included only a single herbivore and a single fungus.
2. The concept and methodology of split-root experiment should be described in more details in Introduction (by adding a diagram?). This would largely simplify the description of subsections 2.3 to 2.7 in the section Materials and methods. By the way, 'Chapter' or 'Chapters' frequently mentioned in the section should be changed Subsection or Subsections.
3. L20, a correction is needed. Beauveria and Metarhizium are classified to Cordycepitaceae and Clavicipitaceae of Hypocreales, respectively.
4. L189-190, 194 and 508 (perhaps more), citations should be coded to match the journal's format.
5. L312, 363 (perhaps more), change per pupae to per pupa.
6. In Figure 1, light versus dark grey error bars are hardly distinguished. Why don't you use different colors to distinguish the treatments? Indeed, the presentation of dark error bars (standard deviations preferred) on distributed data points (different colors and symbols) of all replicates would help to improve the chart quality and to judge statistics in eyeball. The same is true for other figures
Reviewer 3 Report
Posada-Vergara et al. describe studies of entomopathogenic fungi associated with plants, with objectives including the clarification of mechanisms underlying the pathogens' plant-mediated effects on insect herbivory.
This is a thorough (at times meticulously detailed in the methods) study and a clear documentation, accompanied by enough details to warrant confidence in reproducibility. The authors are clear in terms of rationale and justification. The results presented are several and dense, informative and of interest to those working on the interactions of disease and herbivory through plant-mediated effects.
I learned from reading the manuscript and expect it to be of general interest to JoF readers. I have only editorial comments:
L16 "partially caused significant mortality" should be revised to clarify what is meant by partial cause. In association with other causes? The rest of this sentence is also diffuse in terms of suggesting cause. Given that the manuscript is about improving understanding of mechanisms, I suggest you make the statement of causal influence clear here at the beginning of the document.
L23 suggest clarifying how "direct infection" benefits plant health. I assume you mean that direct infection is the route through which the pathogen associates with the plant and conditions for improved response to subsequent challenge.
L78 misplaced "the"
The introduction is very good.
L136 variables, not parameters, are measured; parameters are estimated.
L154 fine camel hair brush (not "camel hairbrush")
L161 the early presentation of these equations is appreciated. I assume that you have data that can be used to address a potential issue with the "missing larvae" component of this equation -- hatching rates or similar. You describe Abbott's correction so it's assumed you're using such data. A sentence mentioning how you recorded hatching under control conditions would be useful here.
L182 "chapter 2.3" reference may be academic-document residue. Update to "section" or something appropriate for an article.
L210 italics formatting of species names
L241 again, "chapter" reference, update to section or similar
L237 genus abbreviation
L244 "chapter" reference -- I'll stop pointing these out, suggesting/assuming that they should be updated to "section" throughout
L302 reference 43 may be clear with respect to calculating LOD and LOQ based on SD, but it would be useful to mention the calculation details here.
Section 2.8 is well written and shows a thoughtful approach to analysis. L313: homoscedasticity is not desired from data, but from residuals. Ratio of damage should not be normal either in "the data" nor the residuals, because ratio should not be analyzed as normally-distributed. This sentence (~L313) should be checked -- clearly you have considered what response distributions are appropriate for analysis, and you are using appropriate ones for proportion/percent variables, and you have even made adjustments for count data due to overdispersion relative to Poisson response distribution. Adjust this sentence/etc. so that it doesn't suggest you have applied normal theory assumptions to your analysis where you probably haven't.
Figure 1 caption and elsewhere: format your references to "p value" consistently. Case, italics. Nice figures; good to see SE that we can assume come from appropriate analysis given your use of binomial response distribution and beta regression elsewhere.
Figure 2: I assume it's r, not R, given that it's negative. I'm concerned about this figure given the high-leverage data points with DNA greater than 5 units. This isn't a major issue to the paper, but a reader is likely to want discussion as to whether you place much belief in this result as a general one [I return to this comment and note that there isn't much discussion of this, and suggest that you add some discussion of it]. You might consider: 1) is there a threshold for DNA above which the relationship matters for pupal survivorship, and below which it does not? If so then the data may be more informative. Is the relationship linear on the domain of DNA quantity? Maybe it's log-linear as is common in dose-response relationships. This is an interesting result that might be worth some additional thinking.
Figure 4 is dense and informative. You could refer to quartiles instead of 25th and 75th percentiles if you would rather.
Figure 5 is even more dense. It doesn't lend itself to quick ascertainment of generalities, but readers working in this field will consume the resource-rich figure with interest.
L482 I think "inhabit" would be better revised to "associate with" -- endophytic vs. herbivorous interactions are very different.
L508 citation formatting
L529 indent or not
L529 rationale here is clear, thank you.
Round 2
Reviewer 1 Report
No comments for authors